# Combining α-s1 Casozepine and Fluoxetine Treatment with a Behavioral Therapy Improves Symptoms in an Aggressive Dog: An Italian Case Report

**DOI:** 10.3390/vetsci10070435

**Published:** 2023-07-04

**Authors:** Luigi Sacchettino, Viviana Orsola Giuliano, Luigi Avallone, Francesco Napolitano, Danila d’Angelo

**Affiliations:** 1Department of Veterinary Medicine and Animal Production, University of Naples Federico II, 80137 Naples, Italy; luigi.sacchettino@unina.it (L.S.); avallone@unina.it (L.A.); danila.dangelo@unina.it (D.d.); 2Independent Researcher, 81100 Caserta, Italy; vetvivianagiuliano@gmail.com; 3CEINGE—Biotecnologie Avanzate Franco Salvatore, 80145 Naples, Italy

**Keywords:** dog behavior, interspecific aggressiveness, human–dog relationship, interdisciplinary approach, nutraceuticals, dog training, SSRI

## Abstract

**Simple Summary:**

Aggressiveness in dogs is referred to as any threatening or harmful behavior toward other animals or humans. It can be triggered by several factors, including illness, anxiety, pain, frustration, fear, genetics, and social and homely settings. Patients affected by aggressive behavior generally experience this kind of psychiatric disorder via different actions, namely positioning the body in a dominant stance, showing teeth, barking, growling, lunging, snapping, snarling, or even biting. Here, we examined the case study of a pure-bred Lagotto Romagnolo, a 4-year-old neutered male dog, who repeatedly attacked one of two family owners (the oldest woman). We documented the beneficial effect of an alternative therapeutic approach based on the combination of the nutraceutical compound α-s1 casozepine as an add on to the conventional calming drug, fluoxetine, together with a specific rehabilitation program.

**Abstract:**

Behavioral dysfunctions in dogs represent a critical issue of the human–animal relationship. In particular, aggression can make interspecific coexistence quite complicated within family units, thus exposing all members to greater health risks. In this present study, we documented multiple aggression episodes against one of the two family members caused by a 4-year-old neutered male pure breed Lagotto Romagnolo dog. To minimize impulsivity and anxiety-like behaviors of the patient as much as possible and improve his relationship with the adopting family, we used an interdisciplinary approach, employing specific skilled personnel, including a veterinary behaviorist and a rehabilitating dog instructor. Nine months after fluoxetine treatment (0.8 mg/kg, SID), in combination with oral α-s1 casozepine administration, and behavioral rehabilitation, the owners reported a significant reduction in aggressive events in terms of intensity and frequency. Collectively, our promising data pave the way toward a more detailed characterization of α-s1-casozepine to better evaluate the potential involvement of such a compound in the modulation of aggressive behaviors in dogs affected by relational dysfunctions.

## 1. Introduction

Dogs have, in current years, accounted for their ability to build up mutual relationships with humans and other species as well via communication and cooperative strategies, including herding, dragging, hunting, gestures, and cues, that many people take advantage of in order to improve the quality of life [1]. Owned dogs generally depend entirely on humans for basic needs, such as food, water, and access to mates, thus suggesting the need for deeper knowledge about human–dog interactions to ensure proper inter- and intra-specific wellbeing. In keeping with that, dysfunctional behaviors of dogs, mainly characterized by aggressive traits towards people, dogs, or other species, are regarded as one of the major threats to society [2,3,4]. Episodes of aggression can bring about injuries, transmission of diseases, and hospitalization of either humans or dogs themselves. These dangerous issues may cause owners to experience distress and poor life quality that they have to make the tough decision to turn to euthanasia for their pets or move them to shelters, where they hardly can recover from the pathological state because of the often inappropriate skills of the kennel staff [5,6,7,8,9,10,11]. It is worth underlying that aggression is not considered a diagnosis but rather a symptom [12], so it can be faced in a proper way only when a clear clinical profile has been drawn. Aggressive behaviors can be clustered into several categories. Indeed, depending on their attitudes, patients can keep their distance or constantly mark their territory, display mothering or fearful behaviors, as well as competition between dogs and people. Unfortunately, aggressions from dogs toward owners represent a typical and substantial threat to public health, and the understanding of aggression etiopathology, as well as effective treatments, are nowadays still lacking [7,11,12,13,14,15,16]. Thus, aggressive and violent behaviors might be considered a multifactorial phenomenon, and their expression is regulated by genetic, physiological, evolutionary, environmental, and social factors [17]. The serotonin reuptake inhibitor, fluoxetine, has been long used to treat behavioral disorders in dogs, including aggression, compulsive disorder, and separation anxiety, due to its ability able to restore thought and action impairments, reduce impulsivity, and cause reflection before acting [18]. In this respect, to reduce health risks to the family and improve the wellbeing of the dog, it is necessary to have a diagnosis and prognosis of the behavioral disorder patient and then apply all the possible therapeutic strategies (behavioral, pharmacological, and/or relational) useful to manage the aggression.

Hence, to prevent further aggressions and injuries toward animals or humans, an appropriate treatment schedule in the management of dog aggression should be taken into consideration. In this respect, an accurate medical evaluation and treatment, if any, education of the family, safety recommendations, management changes, behavior modifications, as well as management tools and medication, are mandatory to build up a stable owner–dog relationship. Such a multifaceted approach involves using consistency, rewards, and training products where necessary to gain effective control, alongside lowering the dog’s arousal, anxiety, fear, or impulsivity [19]. The efficacy of fluoxetine treatment, which has been regarded as one of the main therapeutic options to reduce and mitigate the aggression of dogs toward their owners [20], was also documented in humans who participated in both controlled clinical and experimental trials with a history of aggression [21]. Different pharmacological strategies aimed at counteracting aggression in dogs rely on the use of tricyclic antidepressants or benzodiazepines (i.e., diazepam) in combination with rehabilitation therapy. In some cases, dietary and hormonal modification might be useful to alleviate such behavioral dysfunctions [22,23,24,25]. In this respect, we documented the potential beneficial effect of the α-CZP when chronically administered to an adult compulsive and aggressive dog, together with fluoxetine treatment and a tailored behavioral recovery program [18].

In this present case study, we investigated the impact of an integrated approach, based on the nutraceutical GABA agonist treatment, α-CZP, and fluoxetine, together with a rehabilitation program, upon the interspecific aggressive behavior, in a 4-year-old neutered male purebred dog.

## 2. Case History

We reported a case study of a 4-year-old male Lagotto Romagnolo of 21 kg, who was subjected to our behavioral examination since he caused multiple aggressions against the oldest pet owner of the family. The patient was found on the street while some kids were beating him by members of an animal welfare association. The dog was sheltered in a confined space, where he was not allowed to interact with other dogs or people. This social and environmental isolation lasted until he was adopted at 6 months of age. At the time of the behavioral visit, the dog was living in an apartment with a terrace and courtyard, with two family members: the oldest (mom), affected by walking problems, and the 50-year-old daughter. The owners reported that the patient suffered from dermatological and digestion disorders, so he was prescribed a mono-protein food by the attending veterinarian. The patient showed fear of thunderstorms and sudden loud noises, as well as hyper-alertness at the sight of dogs or cats in the street or when strangers enter the house. He went out twice a day with the younger owner and one more time with the dog sitter, with whom he had built up a harmonious relationship over time.

Games usually performed by the patient were throwing a ball (predatory, hunting behaviors) and pulling the spring (towing behaviors), towards which he generally exhibited himself as very possessive. The first dog’s aggression episode occurred three months after the adoption when the patient came back from the walk and appeared upset since he met other dogs and cats. The oldest owner approached with her face towards the dog in a high-pitched voice, and the animal bit her hand. The second episode was when she sought to get him out of bed by pushing him. In that case, the dog lunged at her, who suddenly fell on the floor. The last episode took place several months later and just before the visit when returning from a very short grooming session, where the length of the hair was greatly reduced (Figure 1). At that time, the mom tried to caress him, and the patient attacked and bit her hand, so she needed to be taken to the emergency room because of the deep injuries caused by the dog.

All the aggressions seemed to be unstructured without a threat phase (neither barking nor growling) or inhibition. During the behavioral examination at the dog’s home, the patient showed a state of persistent agitation, hyper motricity, panting, and continuous vocalizations, as a plea for attention and food. The skin was red and inflamed. Diagnosis was drawn following anamnestic assessment (physical, behavioral, and neurological evaluations, complete blood count, chemistry, and thyroid profile), according to previous work [26]. Blood tests were all in the appropriate ranges. Sensorium–mental attitude, gait and posture, postural reactions, muscle tone, spinal nerve reflexes, and cranial nerve were physiological upon inspection.

After the diagnosis of anxiety-related aggression, the veterinary behaviorist of our team prescribed him a pharmacological and nutraceutical treatment, together with behavioral rehabilitation. In particular, the patient was given α-s1 casozepine (Zylkene^®^, 30 mg/kg) BID and fluoxetine (0.8 mg/kg) SID for 9 months. In line with our previous pharmacological protocol [18], the dog instructor started the rehabilitation program fifteen days after the treatment. Our goal was to reintegrate the threat sequence in the patient, reduce impulsivity, and improve the welfare, thus enabling him to implement collaborative, affiliate, social, exploratory, and scouting motivations through the gaming activities carried out with both owners. The behavioral therapy sessions were weekly at the beginning (3 months), becoming fortnightly in the middle phase (3 months) at the dog’s home. In the last three months, the owners still carried out the activities and routines learned during the rehabilitation on their own.

The owners were initially given basic instructions to adequately manage the patient, such as (1) identifying all potential aggressive situations (e.g., do not touch the dog when he is on the bed or when he comes back from a walk if he looks like nervous) and to avoid threatening situations (e.g., scold the dog in a calm tone to get him out of bed); (2) training in reading canine body language in order to predict and/or prevent bites; (3) segregating the dog when aggression-eliciting stimuli were present, deflecting his attention with chewing games, such as kong; (4) keeping the muzzle if avoidance cannot be ensured (e.g., during medical check or grooming); (5) considering either scruff shaking or alpha rolls as physical/psychological harmful punishment and aggressive techniques since they may encourage avoidance behavior and hostility.

Several follow-up sessions were carried out by the veterinary behaviorist during the therapeutic treatment. The patient showed a sensitive improvement in his aggressive behavior in terms of frequency and intensity seven months after the scheduled therapy, and the owners already reported easier management of the dog (Table 1).

## 3. Discussion

In this present study, we sought to investigate the potential impact of the interdisciplinary and integrated approach based on α-CZP administration as an add on to fluoxetine treatment, together with a tailored rehabilitation program, upon the aggressive behavior of a purebred dog diagnosed with anxiety-related aggression. Previous research reported that dogs suffering from anxiety could become irritable and aggressive, thus requiring pharmacological treatments and behavioral programs [27]. In addition, pain or fear of pain can trigger irritable aggression, which generally arises from approaching or handling [19]. Indeed, this reactivity in dogs can be triggered by different settings, including patting, grooming, drug delivery, massaging a painful part of the body, as well as pulling the dog’s ear or stepping on the tail. Moreover, the dog’s reaction can become sensitized, being exhibited when he anticipates how it will be touched. In fact, fear responses in animals develop when exposed to events or stimuli that are perceived as negative and salient. At first, animals try to show adaptive responses, which are not effective over time. Later on, when they become sensitized to a particular stimulus perceived as a threat, dogs attempt to both identify predictors of that stimulus and develop a response to avoid it [28]. In many cases, the combination of behavioral and medical issues contributes to the onset of aggression in dogs, thus causing the threshold for aggressiveness to be even lower [16,29,30,31]. In the patient of this present study, we failed to find any alterations in the blood, thyroid, and central nervous system functionalities, thus suggesting that aggressions towards the oldest owner were most likely related to an anxiety state since she was affected by alterations in gait and posture, which made the communication ambiguous and threatening as well. Therefore, the choice of combining the nine-month α-CZP and fluoxetine treatment, together with a rehabilitation program, allowed us to achieve a noticeable improvement in impulsivity and aggressive behavior in the patient, who experienced an evident reduction in the general anxiety state and the proper sequence (high stare, growling, barking, snarling, lunging, and snapping) of aggression [32,33,34]. Moreover, during the behavioral examination, the owner reported that the patient suffered from dermatological disorders for which he was prescribed monoproteic nutrition by the attending veterinarian. In this respect, fluoxetine seems to also have some beneficial effects in relieving inflammatory itching and/or pain [35,36], although it is not generally prescribed to treat canine atopic itch [37]. On the other hand, given its ability to bind to GABA_A_ receptors, we administered α-CZP to cope with the anxiety behavior of the patient. Our data point toward the importance of using such a nutraceutical compound as an add on to fluoxetine in order to modulate multiple neurotransmission pathways underlying challenging and complex cases [18,38]. A pioneering study by Beata and colleagues documented a comparable ameliorative effect of both selegiline and α-CZP treatment in anxious dogs following 56-day therapies, suggesting that such a biological compound might be considered a valuable alternative strategy to pharmacological intervention for such a mood-related disorder [39]. Given the functional similarities with benzodiazepine-like activity, the promising role of α-CZP in alleviating physical and psychological anxiety, as well as convulsions, have already been characterized in different clinical and preclinical settings [40,41]. Of note, α-CZP exerts beneficial actions without bringing about adverse effects, found with the therapeutic use of benzodiazepines, among them lethargy, lack of coordination, depression, and cardiovascular or respiratory depression [19]. However, although the precise mechanism of action is still unknown, α-CZP can penetrate the blood–brain barrier once absorbed via the gastrointestinal tract and exert its “calming” effect by stimulating serotonin, dopamine, and GABA receptors [42]. Further studies aimed at better characterizing the downstream molecular pathways modulated via α-CZP need to be carried out. In our recent findings, we documented that the synergistic and complementary role of both fluoxetine and α-CZP, along with cooperative and olfactory tasks, reduced arousal or impulsivity of dogs, who then took pleasure in living with their owners [11,18]. During the behavioral program to treat aggression in dogs, relapse phenomena may appear again, which is why dog owners are strongly advised not to stop the therapy suddenly [43]. The patient we followed acquired so many traumatic, painful experiences, in terms of interaction with humans (the patient was found on the street and even beaten by kids who roamed around), that he had implemented strategies to save him from potential painful contacts. Pain is not always a trigger of the behavioral disorder, but it can exacerbate it [16]. However, we did not consider prescribing painkillers, such as gabapentin or pregabalin, since aggression was linked to a state of anxiety, toward which the gabapentinoides would have only a sedative effect, thus making it very difficult to carry out behavioral rehabilitation in that condition. In addition, studies collected so far about the anxiolytic properties of these compounds are not yet fully understood [44]. Despite the aggression episodes, the severity and frequency of the attacks were greatly reduced, allowing the family to have time enough to be aware of the situation, move away, and minimize the aggressions to an acceptable level for the owners. Collectively, the patient did not experience any recurrence, even 3 months after the end of the treatment. The success of the integrated and interdisciplinary approach was assessed by the veterinarian behaviorist and the owners.

## 4. Conclusions

In this present case report, we sought to investigate the beneficial effect of the nine-month-lasting program based on an integrated and interdisciplinary approach in a clinical case of Lagotto Romagnolo suffering from interspecific aggressive behavior. At the end of the pharmacological treatment with fluoxetine and α-CZP, implemented with a behavioral recovery program, the owners reported a significant reduction in aggression events in terms of intensity and frequency. Our proof-of-concept study might pave the way for the clinical use of such a nutraceutical compound for a long-term regimen as an add-on to conventional therapy to deal with aggression and ensure the wellbeing of the pets. Preclinical in vivo studies in animal models aimed at dissecting the role of GABA and serotonin and identifying their downstream targets involved in dysfunctional behaviors associated with aggressiveness are mandatory. We recognize that case reports do not provide definitive evidence, but they are useful for highlighting observations of importance to the profession.

## Figures and Tables

**Figure 1 vetsci-10-00435-f001:**
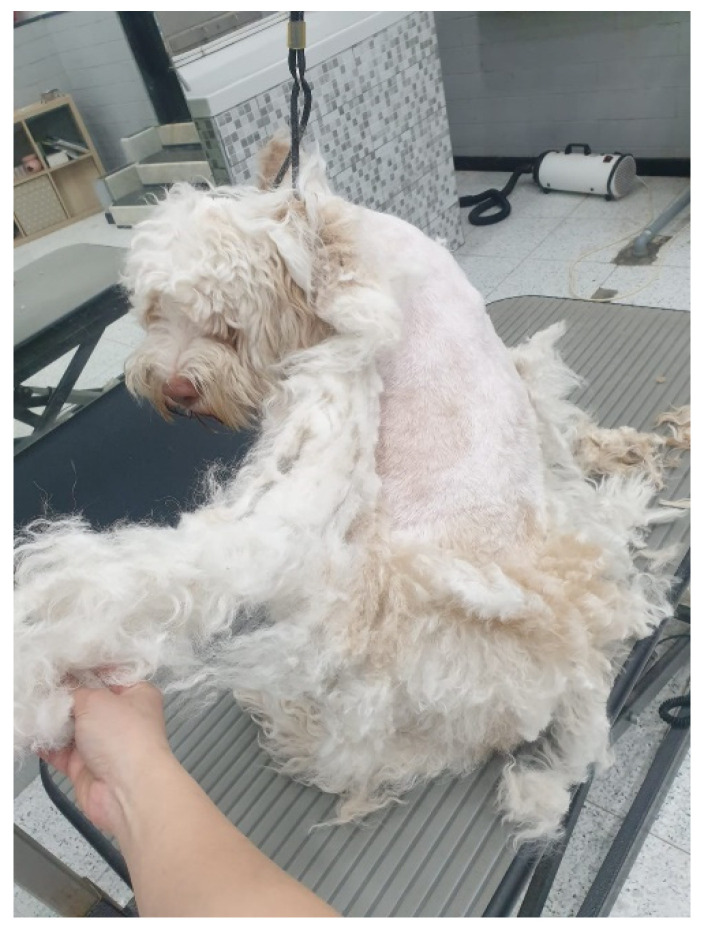
The dog following a very short grooming session.

**Table 1 vetsci-10-00435-t001:** Timeline between the period of follow up and dog’s behavior.

Follow up by Vet Behaviorist	Period	Way	Dog’s Behavior
1°	Fifteen days following pharmacological plus nutraceutical treatment	By phone	Slight inappetence, starting with behavioral rehabilitation therapy.
2°	One month from the rehabilitation program	Behavioral examination	Good response to the treatment, no symptoms of aggressiveness.
3°	Three months from the rehabilitation program	Behavioral examination	First episode of relapse against elderly woman. The dog growled and pushed elderly owner with his paws.
4°	Five months from the rehabilitation program	Behavioral examination	Second episode of relapse against elderly woman; the dog growled and barked.
5°	Seven months from the rehabilitation program	By phone	Good response to the treatment, no relapse from the previous follow up. Dog is more collaborative with all owners.
6°	Eight months from the rehabilitation program	By phone	Stable disease. No symptoms of aggressiveness from the previous follow up. We started reducing the dose of the drug.
7°	Nine months after starting the rehabilitation program	Behavioral examination	Patient was in remission. Pharmacological treatment was stopped. Easier dog management was reported.

## Data Availability

Data is contained within the article.

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
