# Peer review of "Combining α-s1 Casozepine and Fluoxetine Treatment with a Behavioral Therapy Improves Symptoms in an Aggressive Dog: An Italian Case Report"

_vetsci, 2023, doi:10.3390/vetsci10070435_

Round 1

Reviewer 1 Report

REV. Combining α-s1 casozepine and fluoxetine treatment with a behavioral therapy improves symptoms in an aggressive dog: an Italian case report.

The paper explored a case report of a 4-year-old neutered male, pure breed, of Lagotto Romagnolo, who showed multiple aggressions against the eldest of the two owners. The aim of authors was to document the beneficial effect of an alternative treatment using a combination of nutraceutical compound (α-s1 casozepine) and fluoxetine, which is the drug of choice in aggressive behaviour. New knowledge on pharmacological treatments is important in behavioural medicine especially for canine aggressivity.

For this reason, the paper is interesting but some work is needed to improve the manuscript to be suitable for publication in this journal.

Please, consider the following comments.

LL 29 please specify the meaning of low dosage and why the authors have decided for this dosage (see also LL 144)

LL 54-56 there are more canine aggressivity classifications depending on motivations, targets, etc. I suggest to specify this aspect and explain what classification the authors have considered for this case. Add also the references.

LL 63-65 add references

LL 81-102 this part sounds confusing. It is not totally clear what the authors means to point out. I suggest to rephrase organizing better this part. It could be useful describing the usefulness of fluoxetine for the treatment of aggressivity in dog with pro and contra (the classic approach to this behavioural problems) and then explain the benefits of nutraceutical medication.

LL 99-101 add reference

LL 128 explain better what very short grooming mean. The duration or the length of the trimmed hair? And it could be useful adding information on the motivation of this intervention.

LL 136 I understood that during behavioural examination a skin problem was observed. And then a clinical examination was performed. Is it correct?

LL 141 I don’t totally agree with this diagnosis reading the entire manuscript. See LL 230. The authors affirmed that there was a pain condition and the aggression manifestations of dog were more related to a pain-induced aggression. I suggest to clarify better this crucial aspect. Also, it is important to specify what kind of anxiety disorder was diagnosed (maybe generalized anxiety according to the symptoms observed).

LL 178-181 at this point, I suggest to explain that also previous experience and/or traumas can affect the behavioural response of the dogs

LL 193-196 considering the case, it can speculated that the aggression could be an anxiety-related behaviour so it could be possible speak of anxiety-related aggression. It could be relevant to improve the manuscript to describe the anxiety in dog.

LL 196-199 this part should be removed the reference cited is on hereditary eye disease in Lagotto breed and not on the inclination to experience aggressiveness or anxiety. Otherwise please specify better this point.

LL 230-236 as said before, I suggest to reconsider the diagnosis and explain better this part also adding possible differential diagnosis.

Conclusions need to be revised according to the revisions and suggestions. It is very important to add the limitations of this study because is not so clearly demonstrated the benefice effect of using the nutraceutical medication because it was administered in association to fluoxetine. It is not easy to discriminate the role played by alpha-CZP in the treatment of aggression and anxiety in the dog involved in the study. This aspect must be undelined!

LL 257 also and anxiety-related disorder was diagnosed

LL 263-265 this part seems disconnected to the rest of the text. Please clarify better this part or maybe remove it.

Author Response

REV. Combining α-s1 casozepine and fluoxetine treatment with a behavioral therapy improves symptoms in an aggressive dog: an Italian case report.

The paper explored a case report of a 4-year-old neutered male, pure breed, of Lagotto Romagnolo, who showed multiple aggressions against the eldest of the two owners. The aim of authors was to document the beneficial effect of an alternative treatment using a combination of nutraceutical compound (α-s1 casozepine) and fluoxetine, which is the drug of choice in aggressive behaviour. New knowledge on pharmacological treatments is important in behavioural medicine especially for canine aggressivity.

For this reason, the paper is interesting but some work is needed to improve the manuscript to be suitable for publication in this journal.

Please, consider the following comments.

Q: LL 29 please specify the meaning of low dosage and why the authors have decided for this dosage (see also LL 144)

R: We thank the reviewer for her/his comment. In our behavioral experience at University Teaching Hospital we got excellent results by administering fluoxetine at a lower dosage than 1 mg/kg, associating it with a-CZP, bringing about just a transient lack of appetite in patients, as reported by the owners, who are actually not discouraged in using such a compound. The therapeutic efficacy of the “atypical” dose of fluoxetine has also been reported in a previous work by Chutter M and colleagues (2019; doi.org/10.1016/j.jveb.2019.05.006), who documented that the combination of fluoxetine (0.5-1.49 mg/kg once daily) with trazodone or clonidine concurrently produced the highest percentage of positive responses in dogs suffering from fear aggression or generalized anxiety.

Q: LL 54-56 there are more canine aggressivity classifications depending on motivations, targets, etc. I suggest to specify this aspect and explain what classification the authors have considered for this case. Add also the references.

R: We thank the reviewer for the suggestion. We agree with him/her and rewrote the indicated sentence.

Q: LL 63-65 add references

R: Done it.

Q: LL 81-102 this part sounds confusing. It is not totally clear what the authors means to point out. I suggest to rephrase organizing better this part. It could be useful describing the usefulness of fluoxetine for the treatment of aggressivity in dog with pro and contra (the classic approach to this behavioural problems) and then explain the benefits of nutraceutical medication.

R: We thank the reviewer for her/his comment. We rewrote the indicated sentence.

Q: LL 99-101 add reference

R: We thank the reviewer for her/his remark. We moved this sentence, in accord to a previuos comment.

Q: LL 128 explain better what very short grooming mean. The duration or the length of the trimmed hair? And it could be useful adding information on the motivation of this intervention.

R: We thank the reviewer for the remark. In the revised version of the manuscript, we rewrote the indicated sentence. The length of the hair has been greatly reduced during that session, because of a mistake by the groomer. The dog was groomed monthly, as a regular practice of hair management.

Q: LL 136 I understood that during behavioural examination a skin problem was observed. And then a clinical examination was performed. Is it correct?

R: Yes, it is.

Q: LL 141 I don’t totally agree with this diagnosis reading the entire manuscript. See LL 230. The authors affirmed that there was a pain condition and the aggression manifestations of dog were more related to a pain-induced aggression. I suggest to clarify better this crucial aspect. Also, it is important to specify what kind of anxiety disorder was diagnosed (maybe generalized anxiety according to the symptoms observed).

R: We agree with the reviewer upon her/his comment. The dog suffered from a state of generalized anxiety and a remarkable impulsivity, that brought about aggression towards one of the owners. In keeping with with Pineda S. and colleague’s view (2014; doi.org/10.1016/j.tvjl.2013.11.021), who stated that anxious dogs can turn into irritation and aggression, we guess that pain (perceived or presumed) in our patient was a co-morbidity, a worsening factor to trigger pathological behavior. In the revised version of the manuscript, we rewrote the indicated sentence.

Q: LL 178-181 at this point, I suggest to explain that also previous experience and/or traumas can affect the behavioural response of the dogs

R: We thank the reviewer for the advice. We clarified the concept that our patient was acting in a biased manner towards a stimulus that he had coded as threatening.

Q: LL 193-196 considering the case, it can speculated that the aggression could be an anxiety-related behaviour so it could be possible speak of anxiety-related aggression. It could be relevant to improve the manuscript to describe the anxiety in dog.

R: We thank the reviewer for her/his suggestions. In the revised version of the manuscript we better specified the diagnosis, so as to make the text more understandable to the reader.

Q: LL 196-199 this part should be removed the reference cited is on hereditary eye disease in Lagotto breed and not on the inclination to experience aggressiveness or anxiety. Otherwise please specify better this point.

R: We removed it.

Q: LL 230-236 as said before, I suggest to reconsider the diagnosis and explain better this part also adding possible differential diagnosis.

R: Done it.

Q: Conclusions need to be revised according to the revisions and suggestions. It is very important to add the limitations of this study because is not so clearly demonstrated the benefice effect of using the nutraceutical medication because it was administered in association to fluoxetine. It is not easy to discriminate the role played by alpha-CZP in the treatment of aggression and anxiety in the dog involved in the study. This aspect must be undelined!

R: We thank the reviewer for her/his comment, and we amended the conclusions as suggested, in the revised version of the manuscript. We understand that, in an integrated approach, namely a-CZP/fluoxetine and interdisciplinary (vet behaviorist + dog instructor), it is not easy to estimate and dissect the therapeutic efficacy of the individual active ingredient, that’s why we think that preclinical in vivo studies might disclose this issue. The patient of our study showed an anxiety-related aggression, and this was the reason why our therapeutic choice fell on the nutraceutical compound a-CZP, the anxiolytic-like compound able to bind GABAA receptors. Therefore, as also mentioned before, our idea was to target serotonin and GABA neurotransmissions in a “safer” way, thus trying to manage and offset the potential side effects of either, when modulated at pharmacological level (i.e., higher fluoxetine dosage and gabapentin administration). On top of that, taking advantage of a detailed behavioral program, we worked on the social environment in which the dog lived, that was re-shaped in order to make it more predictable and satisfying from an ethological and wellbeing point of view.

Q: LL 257 also and anxiety-related disorder was diagnosed

R: Done it.

Q: LL 263-265 this part seems disconnected to the rest of the text. Please clarify better this part or maybe remove it.

R: We removed it in the revised version of the manuscript.

Reviewer 2 Report

About the paper presentation:

The paper has interesting data but will gain in a better presentation and constructions: sections such as signalment, demographic (e.g., history of the patient), medical and behavioral history, physical and behavioral signs, diagnosis (differential and final), treatment, follow up and discussion will help to better present and prioritize the elements of this case. In the actual form it is difficult to follow the author clinical construction.

About the diagnosis:

It will gain to be more precise and justified (see remarks of line 141). The behavior of the dog during the consultation, particularly continuous vocalizations in request for attention and food is not enough explained and exploited. ANd it could be many other options and the clinical reasoning is clearly lacking

About the treatment:

The choice to give 0,8 mg/kg of fluoxetine to avoid side effect is not based no peer reviewed reference, e.g., nothing tells us that 0.8 mg/kg will help to avoid known side effect. Actually, slight inappetence is reported by the owners.Many references describe either doses with not more side effects (e.g. Bleuer-elsner et al, 2021)

Behavioral modifications focus on good guideline to improve owner’s security and avoid positive punishments, but no desensitization or counter conditioning method have been proposed. It should be discussed.

About the discussion:

Some points are not discussed.

Since fluoxetine reduce impulsivity, aggressions and pain on its own it is difficult to know if the addition of α-s1 casozepine is really an advantage. Absolutely nothing in the design (like interrupting alpha casozepine) shows that the improvement would not have been the same without alpha casozepine. Also, several papers show the absence of proof for the efficacy of alpha casozepine and are not discussed int he case.

Furthermore, the behavioral modifications proposed in the paper may be enough to ensure the security and the avoidance of aggressive behaviors.

As pointed above, desensitization or counter conditioning method have been proposed. It should be discussed.

Pain and emotional conditioning linked to care are the causes of the dog’s behavior. Pain killers or other molecules with action on pain and anxiety such as gabapentinoïdes should be discussed

Specific comments

Line 41 to 44

It would be nice to cite the article of Wynne

Wynne, C.D.L., 2021. The Indispensable Dog. Front. Psychol. 12, 656529. https://doi.org/10.3389/fpsyg.2021.656529

Line 48

“These unpleasant issues may cause distress in owners, who very often experience a poor life quality”

Unpleasant is an understatement. The dog can represent a real danger for the owners and strangers

I would change “unpleasant” for “dangerous”

Lines 52 to 54

“Worth underlying that aggressiveness is not a diagnosis as a such [11], so it can be faced in a proper way only when a clear clinical profile has been drawn.”

I suggest to complete by” … aggressiveness is not  a diagnosis as a such but a symptom, so it can be faced in a proper way only when a clear diagnosis (instead of clinical profile) has been drawn.”

Lines 54 to 56

“Aggressive behavior can be categorized into defensive, distancing, territorial, maternal, irritable, fearful, displaced, competitive between dogs and people, as well as possessive between dogs belonging to different social groups, hunting, excessive on command.”

References on this classification is needed.

Line 61 to 63

“Aggressive behaviors are generally associated to the functional imbalance between cortical control and the downstream activation of limbic areas, such as amygdala, frontal and temporal lobe, which might impact upon monoaminergic and catecholaminergic neurotransmission.”

         Some remarks on this sentence. The main top-down regulation is emitted from the prefrontal cortex

(see. Carlson, N.R., Birkett, M.A., 2017. Physiology of behavior, Twelfth edition, global edition. ed, Always learning. Pearson, Boston Columbus Indianapolis New York City San Francisco Amsterdam Cape Town Dubai London.

Stahl, S.M., Muntner, N., 2021. Stahl’s essential psychopharmacology: neuroscientific basis and practical applications, Fifth edition. ed, Medicine. Cambridge University Press, Cambridge, United Kingdom New York Melbourne New Delhi Singapore.)

Catecholamine includes monoamines: noradrenaline, adrenaline and dopamine. The sentence should be rewrite I suggest simply “…which might impact upon monoaminergic neurotransmission.”

Aggressive behavior implies the gray aqueductal gray matter that is not mentioned here.

Line 77-78

Of note, the N-terminal decapeptide -casozepine (-CZP), obtained from the tryptic hydrolysis of bovine aS1-casein, can bind to GABAA receptors at the benzodiazepine site ..”

I will be more precise: In vitro α-casozepine shows an affinity ten thousand times weaker than diazepam for GABA-A, but it is ten times more effective than the latter in the paradigm of conditioned defensive burial in rats ( burying reflex in the face of danger) (Schroeder et al., 2003; Miclo, 2001).

Line 85-85

In order to prevent further aggressions and injuries, an appropriate treatment schedule in the management of dog aggression should take into consideration.

Check the English

Line 87 to 89

In this respect, medical evaluation, and treatment as necessary, education of the family, safety recommendations, management changes, behavior modifications, as well as management tools and medication, are mandatory to build up a stable owner-dog relationship.

Wording is unclear. What difference does the author make between treatment and all the rest of the sentence? Difference between management changes and behavior modifications ?

Line 90 to 93

Such a multifaced approach takes place using consistency, rewards, and training products where necessary to gain effective control, alongside with lowering ..”

Check the English

Line 141

After the diagnosis of irritable aggressiveness and anxiety disorder…”

The author mention at the beginning of the paper “aggressiveness is not  a diagnosis as a such” and here it appears to be one. Can you precise ? I would prefer irritable aggression as a symptom and anxiety disorder as a diagnosis.

There is no systematic differential diagnosis

The irritable aggression is not explained, irritated by what ? The skin ?

Anxiety is not explained or defined: intermittent? General ? What is the etiology in this case? Can we emit hypothesis?

Line 172

“diagnosed as a kind of behaviors correlated to irritable aggression and anxiety disorder.”

Again a mix between symptom and diagnosis. Define better the type of anxiety

Line 173 to 176

In line with Landsberg and colleagues, irritable aggression is often referred to as …”

Check the English, I guess the author means that pain or the fear to feel pain can trigger irritable aggression.

Line 178-179

Check the English

Line 197 to 199

Among the susceptibility factors for some race-dependent disorders,   the Lagotto Romagnolo seemed not to have the inclination to experience either aggressiveness or anxiety

The author refers to a paper “A multicenter retrospective evaluation of the prevalence of known and presumed hereditary eye diseases in Lagotto Romagnolo dog breed within a referral population in Italy

I don’t recall any validated research on dog breed inclination for aggressiveness or anxiety

Line 199

Therefore, the nine-month …”

Taking in account the precedent remark the causality is not so obvious as the author likes to state

Line 203

Moreover, the patient started suffering from dermatitis and itching, because the coat was cut very short; thus, we prescribed fluoxetine, aiming at counteracting also such a troublesome symptom

Fluoxetine can’t be considered as a first line treatment for dermatitis and itching. And it still unclear and not debated if the skin inflammation wasn’t a part of the differential diagnosis.

Line 242-245

since pain can affect mood-related activities, characterized by a reduced brain serotonin content

The article in references : Mills, D.S.; Demontigny-Bedard, I.; Gruen, M.; Klinck, M.P.; McPeake, K.J.; Barcelos, A.M.; Hewison, L.; Van Haevermaet, H.; Denenberg, S.; Hauser, H.; et al. Pain and Problem Behavior in Cats and Dogs. Animals (Basel) 2020, 10, doi:10.3390/ani10020318

Does not support the choice of fluoxetine.

Author Response

About the paper presentation:

The paper has interesting data but will gain in a better presentation and constructions: sections such as signalment, demographic (e.g., history of the patient), medical and behavioral history, physical and behavioral signs, diagnosis (differential and final), treatment, follow up and discussion will help to better present and prioritize the elements of this case. In the actual form it is difficult to follow the author clinical construction.

About the diagnosis:

Q: It will gain to be more precise and justified (see remarks of line 141). The behavior of the dog during the consultation, particularly continuous vocalizations in request for attention and food is not enough explained and exploited. ANd it could be many other options and the clinical reasoning is clearly lacking

R: We thank the reviewer for her/his remark. In the revised version of the manuscript we better addressed this point about diagnosis.

About the treatment:

Q: The choice to give 0,8 mg/kg of fluoxetine to avoid side effect is not based no peer reviewed reference, e.g., nothing tells us that 0.8 mg/kg will help to avoid known side effect. Actually, slight inappetence is reported by the owners.Many references describe either doses with not more side effects (e.g. Bleuer-elsner et al, 2021).

R: We thank the reviewer for her/his comment. In our behavioral experience at University Teaching Hospital we got excellent results by administering fluoxetine at a lower dosage than 1 mg/kg, associating it with a-CZP, bringing about just a transient lack of appetite in patients, as reported by the owners, who are actually not discouraged in using such a compound. The therapeutic efficacy of the “atypical” dose of fluoxetine has also been reported in a previous work by Chutter M and colleagues (2019; doi.org/10.1016/j.jveb.2019.05.006), who documented that the combination of fluoxetine (0.5-1.49 mg/kg once daily) with trazodone or clonidine concurrently produced the highest percentage of positive responses in dogs suffering from fear aggression or generalized anxiety.

Q: Behavioral modifications focus on good guideline to improve owner’s security and avoid positive punishments, but no desensitization or counter conditioning method have been proposed. It should be discussed.

R: We thank the referee for her/his comment. Our approach to the rehabilitation of behavior is cognitive-relational. We worked on the family system, to ensure that dysfunctional behaviors could be more manageable, and let those proposals aiming at ensuring the ethology of the patient came out. Thus, we introduced olfactory and gaming activities of both dog and owner, so as to reduce impulsiveness, make interactions more predictable and pleasant (reducing the cognitive bias of the dog towards the human, experienced as threatening). We did not report in detail the rehabilitation protocol for the sake of the manuscript length limits.

About the discussion:

Some points are not discussed.

Q: Since fluoxetine reduce impulsivity, aggressions and pain on its own it is difficult to know if the addition of α-s1 casozepine is really an advantage. Absolutely nothing in the design (like interrupting alpha casozepine) shows that the improvement would not have been the same without alpha casozepine. Also, several papers show the absence of proof for the efficacy of alpha casozepine and are not discussed int he case.

R: We thank the referee for her/his comment and totally agree upon this issue. As also stated in the discussion of the manuscript, in an integrated approach, namely a-CZP/fluoxetine and interdisciplinary by vet behaviorist plus dog instructor, it is not easy to estimate and dissect the therapeutic efficacy of the individual active ingredient, that’s why we think that preclinical in vivo studies might disclose this issue. To make more reliable this approach further studies on a wider cohort of patients are mandatory. In a proposal research project, which we’ve applied for, we proposed to dissect the potential role of a-CZP and fluoxetine (administered alone or combination and at different concentrations), upon behavioral, brain connectivity (fMRI), and electrophysiological phenotypes, in a mouse model of mood-related disorders. Unfortunately, we couldn’t carry out what indicated by the reviewer, mainly because of ethical concerns and owners’ collaboration as well, which would certainly represent not negligible pitfalls. However, the present clinical report should be considered a proof-of-concept study, pointing out the beneficial effects of such an integrated approach, as a sort of alternative therapy to the conventional treatments.

Q: Furthermore, the behavioral modifications proposed in the paper may be enough to ensure the security and the avoidance of aggressive behaviors. As pointed above, desensitization or counter conditioning method have been proposed. It should be discussed.

R: We thank the reviewer for her/his comment and understand the point of view. In the cognitive-relational approach there is a lot of importance to the communication between dog and owner. We worked on the family system, to ensure that dysfunctional behaviors could be more manageable, and let those proposals aiming at ensuring the ethology of the patient came out. Thus, we introduced olfactory and gaming activities of both dog and owner, to reduce impulsiveness, make interactions more predictable and pleasant (reducing the cognitive bias of the dog towards the human, experienced as threatening). During the desensitization procedure, the patient is generally exposed to several anxiety-inducing situations, and attempts are made by him to induce an immediate reaction, which is inconsistent with anxiety-related response. Our approach was instead based on the idea of modifying the dysfunctional representation that the dog had, about the human, experienced as threatening.

Q: Pain and emotional conditioning linked to care are the causes of the dog’s behavior. Pain killers or other molecules with action on pain and anxiety such as gabapentinoïdes should be discussed

R: We thank the referee for her/his comment and totally agree upon this. In the revised versione of the manuscript we implemented the discussion as suggested.

Specific comments

Q: Line 41 to 44 : It would be nice to cite the article of Wynne

Wynne, C.D.L., 2021. The Indispensable Dog. Front. Psychol. 12, 656529. https://doi.org/10.3389/fpsyg.2021.656529

R: We really appreciated the reviewer’s advice, and proceeded to add the reference in the revised version of the manuscript.

Q: Line 48

“These unpleasant issues may cause distress in owners, who very often experience a poor life quality”

Unpleasant is an understatement. The dog can represent a real danger for the owners and strangers

I would change “unpleasant” for “dangerous”

R: Thank you very much for the suggestion. We amended this point accordingly.

 Q: Lines 52 to 54 “Worth underlying that aggressiveness is not a diagnosis as a such [11], so it can be faced in a proper way only when a clear clinical profile has been drawn.” I suggest to complete by” … aggressiveness is not  a diagnosis as a such but a symptom, so it can be faced in a proper way only when a clear diagnosis (instead of clinical profile) has been drawn.”

R: we thank the reviewer for her/his comment. We improved the section, as suggested.

 Q: Lines 54 to 56 “Aggressive behavior can be categorized into defensive, distancing, territorial, maternal, irritable, fearful, displaced, competitive between dogs and people, as well as possessive between dogs belonging to different social groups, hunting, excessive on command.”

References on this classification is needed.

R: Thank you very much for the annotation. In a previous version we inserted the references, that for some reasons we’ve lost in the current version. In the modified version of the manuscript we put the back.

Q: Line 61 to 63 “Aggressive behaviors are generally associated to the functional imbalance between cortical control and the downstream activation of limbic areas, such as amygdala, frontal and temporal lobe, which might impact upon monoaminergic and catecholaminergic neurotransmission.”

Some remarks on this sentence. The main top-down regulation is emitted from the prefrontal cortex

(see. Carlson, N.R., Birkett, M.A., 2017. Physiology of behavior, Twelfth edition, global edition. ed, Always learning. Pearson, Boston Columbus Indianapolis New York City San Francisco Amsterdam Cape Town Dubai London.

Stahl, S.M., Muntner, N., 2021. Stahl’s essential psychopharmacology: neuroscientific basis and practical applications, Fifth edition. ed, Medicine. Cambridge University Press, Cambridge, United Kingdom New York Melbourne New Delhi Singapore.)

Catecholamine includes monoamines: noradrenaline, adrenaline and dopamine. The sentence should be rewrite I suggest simply “…which might impact upon monoaminergic neurotransmission.”

Aggressive behavior implies the gray aqueductal gray matter that is not mentioned here.

R: We thank the reviewer for her/his thoughtful suggestions raised. In the revised version of the manuscript, we mentioned the crucial role of both PFC and PAG brain region in modulating aggressive-related behaviors.

 Q: Line 77-78 “Of note, the N-terminal decapeptide -casozepine (-CZP), obtained from the tryptic hydrolysis of bovine aS1-casein, can bind to GABAA receptors at the benzodiazepine site ..”

I will be more precise: In vitro α-casozepine shows an affinity ten thousand times weaker than diazepam for GABA-A, but it is ten times more effective than the latter in the paradigm of conditioned defensive burial in rats ( burying reflex in the face of danger) (Schroeder et al., 2003; Miclo, 2001).

R: We amended the section, as suggested.

Q: Line 85-85 “In order to prevent further aggressions and injuries, an appropriate treatment schedule in the management of dog aggression should take into consideration.” Check the English

R: Done it.

Q: Line 87 to 89 “In this respect, medical evaluation, and treatment as necessary, education of the family, safety recommendations, management changes, behavior modifications, as well as management tools and medication, are mandatory to build up a stable owner-dog relationship.”

Wording is unclear. What difference does the author make between treatment and all the rest of the sentence? Difference between management changes and behavior modifications ?

R: We rewrote the sentence accordingly, making it more understandable.

Q: Line 90 to 93 “Such a multifaced approach takes place using consistency, rewards, and training products where necessary to gain effective control, alongside with lowering ..” Check the English

R: Done it.

Q: Line 141 “After the diagnosis of irritable aggressiveness and anxiety disorder…” The author mention at the beginning of the paper “aggressiveness is not  a diagnosis as a such” and here it appears to be one. Can you precise ? I would prefer irritable aggression as a symptom and anxiety disorder as a diagnosis.

The irritable aggression is not explained, irritated by what ? The skin ? Anxiety is not explained or defined: intermittent? General ? What is the etiology in this case? Can we emit hypothesis?

R: We agree with the reviewer upon her/his comment. The dog suffered from a state of generalized anxiety and a remarkable impulsivity, that brought about aggression towards one of the owners. In keeping with with Pineda S. and colleague’s view (2014; doi.org/10.1016/j.tvjl.2013.11.021), who stated that anxious dogs can turn into irritation and aggression, we guess that pain (perceived or presumed) in our patient was a co-morbidity, a worsening factor to trigger pathological behavior. In the revised version of the manuscript, we rewrote the indicated sentence.

Q: Line 172 “diagnosed as a kind of behaviors correlated to irritable aggression and anxiety disorder.”

Again a mix between symptom and diagnosis. Define better the type of anxiety

R: Done it.

Q: Line 173 to 176 “In line with Landsberg and colleagues, irritable aggression is often referred to as …”

Check the English, I guess the author means that pain or the fear to feel pain can trigger irritable aggression.

R: We rewrote the sentence.

Q: Line 178-179 Check the English

R: Done it.

Q: Line 197 to 199“ Among the susceptibility factors for some race-dependent disorders,  … the Lagotto Romagnolo seemed not to have the inclination to experience either aggressiveness or anxiety” The author refers to a paper “A multicenter retrospective evaluation of the prevalence of known and presumed hereditary eye diseases in Lagotto Romagnolo dog breed within a referral population in Italy”

I don’t recall any validated research on dog breed inclination for aggressiveness or anxiety.

R: We totally agree with the reviewer’s doubt raised. The idea was to underly the genetic susceptibility of dog breeds to develop a variety of disorders. However, we realized that the sentence as reported is misleading and less clear. Therefore, in the revised version of the manuscript we deleted it accordingly.

 Q: Line 199 “Therefore, the nine-month …” Taking in account the precedent remark the causality is not so obvious as the author likes to state.

R: We really appreciate the remark of the reviewer. However, we reported the data as obtained from the integrated therapy. One of the positive outcomes we had, and that we are quite proud of, is that the owners (especially the oldest one) reported no any further aggressions, even 4 months from the end of therapy, thus suggesting that this combined approach might be promising. Finally, as we stated before, clinical and preclinical studies are needed to better disclose and appraise the functional contribution of a-CZP.

Q: Line 203  “Moreover, the patient started suffering from dermatitis and itching, because the coat was cut very short; thus, we prescribed fluoxetine, aiming at counteracting also such a troublesome symptom” Fluoxetine can’t be considered as a first line treatment for dermatitis and itching. And it still unclear and not debated if the skin inflammation wasn’t a part of the differential diagnosis.

R: We agree with the referee upon this issue. In the revised version of the manuscript, we better clarify this point. In this respect, the patients didn’t take any other medications, during the behavioral therapy, as well as diet was unchanged. Fluoxetine can’t be considered as a first line treatment for dermatitis and itching. However, we chose it in order to counteract impulsiveness and itching/pain.

Q: Line 242-245 “since pain can affect mood-related activities, characterized by a reduced brain serotonin content” The article in references : Mills, D.S.; Demontigny-Bedard, I.; Gruen, M.; Klinck, M.P.; McPeake, K.J.; Barcelos, A.M.; Hewison, L.; Van Haevermaet, H.; Denenberg, S.; Hauser, H.; et al. Pain and Problem Behavior in Cats and Dogs. Animals (Basel) 2020, 10, doi:10.3390/ani10020318.

Does not support the choice of fluoxetine.

R: we’re grateful to the reviewer for the issue raised. We rewrote the sentence in the revised version of the manuscript, thus emphasizing the role of pain as enhancer of the aggression, rather than trigger.

Round 2

Reviewer 1 Report

Rev 2

The manuscrip is improved. The aim of this study is to assess the impact of an integrated approach, based on the nutraceutical GABA agonist, α-CZP, and fluoxetine as a treatment of the interspecific aggressive behaviour.

Noe the aim of the study is more clear and better explained

Please, consider the following comments.

The introduction appears too long, in particular the pharmacological description. Some of this information could be integrate in the discussion.

LL 27 and anxiety-like behaviour

LL 366-367 the impact of an integrated approach, based on the nutraceutical GABA agonist, α-CZP, and fluoxetine, together with a rehabilitation program

LL 799-800 add references

LL 801-803 this sentence is a repetition. LL 719-720

LL 805 they are

LL 811-813 did the authors observe the restore the correct sequence preceding aggression?

LL 815 did not

LL 819-823 it is a repetition of the previous sentence.

LL 834-835 reference

LL 911-920 this part is redundant! authors have already discussed about the role of pain

LL925 reference

LL 928-932 also in this case the part on pain is redundant. There are some other active substances that can control the pain without sedative effects, also natural or nutraceutical compounds!

LL 951 I suggest to remove pain-related behaviors

The discussion is quite unfocused and dispersive, in some parts repetitive. It could be better organized.

Author Response

The manuscrip is improved. The aim of this study is to assess the impact of an integrated approach, based on the nutraceutical GABA agonist, α-CZP, and fluoxetine as a treatment of the interspecific aggressive behaviour.

Noe the aim of the study is more clear and better explained

Please, consider the following comments.

Q: The introduction appears too long, in particular the pharmacological description. Some of this information could be integrate in the discussion.

R: Thank you for this suggestion. We reduced it, according to the comment’s reviewers.

Q:LL 27 and anxiety-like behaviour

R: Done it

Q: LL 799-800 add references

R: Done it

Q: LL 801-803 this sentence is a repetition. LL 719-720

R: We thank the reviewer’s note. In the revised version of the manuscript, we deleted it.

Q: LL 805 they are

R: Done it

Q: LL 811-813 did the authors observe the restore the correct sequence preceding aggression?

R: Yes, we did. We observed that the patient could perform a correct behavioral sequence after the rehabilitation program we followed.

Q: LL 815 did not

R: We thank the reviewer for her/his note and amended it accordingly.

Q: LL 819-823 it is a repetition of the previous sentence.

R: We thank the reviewer’s note. In the revised version of the manuscript, we deleted it.

Q: LL 834-835 reference

R: Done it.

Q: LL 911-920 this part is redundant! authors have already discussed about the role of pain

R: We totally agree with the reviewer for her/his annotation. In the revised version of the manuscript, we deleted it.

Q: LL925 reference

R: We Added it.

Q: LL 928-932 also in this case the part on pain is redundant. There are some other active substances that can control the pain without sedative effects, also natural or nutraceutical compounds!

R: We totally agree with the reviewer upon her/his thoughtful comment. We usually administer painkiller compounds in our therapy: cannabis, cannabinoids, β-Caryophyllene, PEA, boswellia serrata, harpagophytum procumbens, curcumin, quercetin, omega 3. However, based on the previous history of the patients, and according to what suggested by the reviewer in the last revision of the manuscript, we mainly focused on the generalized anxiety (see discussion), which the dog suffered from, as a trigger for the aggression, rather than pain. In addition, we focused on the pain-relieving power of fluoxetine to cover a pain component, reserving painkillers in a second measure, if our therapeutic strategy would not have reserved the effects we wanted. In this case report, our first choice of treatment to counteract aggression was fluoxetine and α-CZP which, together with rehabilitation therapy, proved to be successful, at least in our conditions. Therefore, these case study data might be of interest to carry out detailed preclinical studies, about the potential role of α-CZP, as well as other nutraceuticals, in improving behavioral dysfunctions in dogs.

Q: LL 951 I suggest to remove pain-related behaviors

R: Done it.

Q: The discussion is quite unfocused and dispersive, in some parts repetitive. It could be better organized.

R: We agree with the reviewer for her/his remark. In the revised version of the manuscript we took steps to make the discussion more understandable and less repetitive.

Reviewer 2 Report

I think that the manuscript can be accepted with the changes made by the authors

Author Response

We thank the reviewer for appreciating our revision.